# MLEP: Multi-granularity Local Entropy Patterns for Generalized AI-generated Image Detection

**Lin Yuan, Xiaowan Li, Yan Zhang,* Jiawei Zhang, Hongbo Li, Xinbo Gao***
Chongqing Key Laboratory of Image Cognition,
Chongqing University of Posts and Telecommunications, Chongqing 400065, China
yuanlin@cqupt.edu.cn, s230201063@stu.cqupt.edu.cn,
{yanzhang1991, zhangjw, lihongbo, gaoxb}@cqupt.edu.cn

## Abstract

Advances in image generation technologies have raised growing concerns about their potential misuse, particularly in producing misinformation and deepfakes. This creates an urgent demand for effective methods to detect AI-generated images (AIGIs). While progress has been made, achieving reliable performance across diverse generative models and scenarios remains challenging due to the absence of source-invariant features and the limited generalization of existing approaches. In this study, we investigate the potential of using image entropy as a discriminative cue for AIGI detection and propose Multi-granularity Local Entropy Patterns (MLEP), a set of feature maps computed based on Shannon entropy from shuffled small patches at multiple image scales. MLEP effectively captures pixel dependencies across scales and dimensions while disrupting semantic content, thereby reducing potential content bias. Based on MLEP, we can easily build a robust CNN-based classifier capable of detecting AIGIs with enhanced reliability. Extensive experiments in an open-world setting, involving images synthesized by 32 distinct generative models, demonstrate that our approach achieves substantial improvements over state-of-the-art methods in both accuracy and generalization. Our code and models are available at `https://www.github.com/fkeufss/MLEP/`.

## 1 Introduction

The rapid development of generative technologies has transformed image synthesis, with models like GAN [1], diffusion model [2], and their variants achieving impressive realism. While enabling new applications in creative industries, these advancements have also raised concerns over misuse in misinformation and deepfakes [3, 4], prompting an urgent need for reliable AI-generated image (AIGI) detection methods. Researchers have leveraged spatial [5, 6, 7, 8] and frequency-domain cues [9, 10, 11, 12], as well as high-level knowledge from pretrained diffusion models [13, 14] and LLMs [15, 16, 17] for AIGI detection. Yet, the lack of source-invariant representations still limits the cross-domain detection robustness, especially when across different models and content types.

To address this challenge, we aim to identify a generalized, content-agnostic pattern that can reliably distinguish AIGIs from real photographs. Inspired by recent studies [7, 8], our work builds on two key observations. Tan et al.[7] found that generative models typically involve internal upsampling operations and propose Neighboring Pixel Relationships (NPR) to capture resulting structural artifacts. However, NPR operates on small local patches and retains visible semantic structures, introducing bias that may hinder generalization. Zheng et al.[8] emphasized the impact of "semantic artifacts" on detection and propose disrupting image semantics by shuffling $32 \times 32$ patches. While this improves

---

*Corresponding authors.

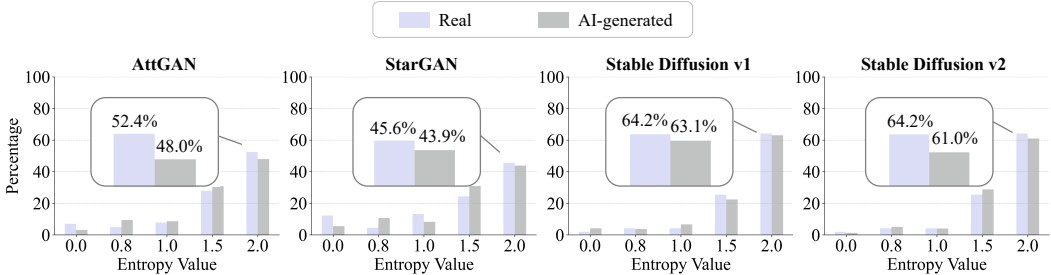

Figure 1: Comparison of local entropy distributions between real and AI-generated images using $2 \times 2$ patches, with entropy values from $\{0, 0.8, 1.0, 1.5, 2.0\}$. Real images consistently show a higher likelihood of entropy reaching 2.0.

cross-scene generalization, the relatively large patch size still preserves semantic information. We argue that such artifacts persist and continue to limit content-agnostic detection.

Through large-scale subjective observation, we noticed a distinct "glossy and smooth" texture in AI-generated images, prompting an investigation into their entropy characteristics, a statistical measure of pixel randomness [18]. We conducted a preliminary study comparing local entropy distributions (using $2 \times 2$ patches) between real and AI-generated images. As shown in Fig. 1, real images consistently exhibit a higher probability of maximum entropy (2.0), suggesting the potential of entropy as a discriminative feature for AIGI detection. Motivated by this, we propose using image entropy as an alternative to pixel differences as proposed by NPR [7]. Entropy captures pixel relationships while reducing semantic dependency by focusing on pixel value distributions rather than contrasts. To further suppress semantic artifacts, we adopt fine-grained patch shuffling (smaller than the $32 \times 32$ patches used in [8]), which also reduces the computational overhead of entropy computation. Additionally, we incorporate multi-scale resampling and an overlapping sliding window to enhance the granularity of entropy patterns. Our contributions are summarized as follows:

- To the best of our knowledge, it is the first attempt to explore the potential of image entropy as a cue for detecting AI-generated images. Using image entropy not only enhances detection accuracy and generalization compared to state-of-the-art methods but also highlights intrinsic differences between real and AI-generated images in terms of pixel randomness, as quantified by image entropy.

- We propose **M**ulti-granularity **L**ocal **E**ntropy **P**atterns (**MLEP**), a set of feature maps with entropy computed from shuffled small patches across multiple resampling scales. MLEP effectively disrupts image semantics to mitigate content bias, while capturing pixel relationships across both spatial and scale dimensions. Using MLEP as input, a standard CNN classifier can be trained for robust and generalized AIGI detection.

- Extensive quantitative and qualitative analyses validate the effectiveness of the MLEP design, showing significant improvements over state-of-the-art methods across multiple AI-generated image datasets.

## 2 Related Work

**Spatial-domain Detection** Spatial-domain methods typically rely on handcrafted spatial features, local patterns, or pixel statistics to distinguish between real and generated images. The representative methods include generalized feature extraction from CNN-based model [5] and inter-pixel correlation between rich and poor texture regions [6]. Tan et al. [7] observed that upsampling operations are prevalent in image generation models and proposed utilizing neighboring pixel relationships (NPR), computed through local pixel differences, as a simple yet effective cue for AIGI detection. Zheng et al. [8] discovered that image semantic information negatively impacts detection performance and proposed a simple linear classifier that utilizes image patch shuffling to disrupt the original semantic artifacts. Cozzolino et al. [19] proposed a zero-shot detection method that models the distribution of real images using lossless coding with multi-resolution prediction, identifying AI-generated images by detecting higher-than-expected coding costs that indicate deviations from real-image

statistics. Yang et al. [20] proposed Discrepancy Deepfake Detector ($D^3$), which enhances cross-generator generalization by introducing a parallel branch that extracts a discrepancy signal from distorted features to complement the original representation, achieving better robustness without compromising in-domain performance.

**Frequency-domain Detection**   To tackle the subtlety of spatial artifacts in AI-generated images, frequency-domain methods analyze image frequency components, enabling more effective real vs. fake differentiation. The study in [9] found that GAN-generated images exhibit generalized artifacts in discrete cosine transform (DCT) spectrum, which can be readily identified. Qian et al. [10] proposed a face forgery detection network based on frequency-aware decomposed image components and local frequency statistics. Luo et al. [11] proposed a feature representation based on high-frequency noise at multiple scales and enhanced detection performance by integrating it with an attention module. Liu et al. [12] utilized noise patterns in the frequency domain as feature representations for detecting AI-generated images. Tan et al. [21] proposed FreqNet toward detection generalizability, which focuses on high-frequency components of images, exploiting high-frequency representation across spatial and channel dimension.

**Detection leveraging Pretrained Models**   This group of methods aims to derive generalized features for AIGI detection by leveraging the knowledge learned by large models pretrained on extensive datasets. Wang et al. [13] proposed an artifact representation named DIffusion Reconstruction Error (DIRE), which obtains the difference between the input image and its reconstructed object through a pretrained diffusion model. Chen et al. [14] proposed utilizing pretrained diffusion models to generate high-quality synthesized images, serving as challenging samples to enhance the detector's performance. Ojha et al. [15] utilized representations from a fixed pretrained CLIP model as generalized features for detection. Chen et al. [22] introduced ForgeLens, a data-efficient CLIP-ViT framework that enhances generalization to unseen forgeries by using proposed lightweight weight shared guidance module (WSGM) and forgery-aware feature integrator (FAFormer) to guide frozen features toward forgery-relevant information. Zhang et al. [23] proposed VIB-Net, which employs variational information bottlenecks to enforce authentication task-related feature learned from pretrained CLIP encoder, significantly improving generalization across different generative model types. Similar methods such as [16, 17] also leveraged textual information from vision-language models to further enhance detection performance.

## 3   The Approach

The proposed approach leverages entropy-based feature extraction to analyze local pixel randomness in a multi-granularity, semantic-agnostic manner. It begins by dividing the image into small patches and applying random shuffling to reduce semantic bias. A multi-scale pyramid is then constructed by downsampling and upsampling the scrambled image, introducing resampling artifacts. Local entropy is computed using a $2 \times 2$ sliding window across the entire image, capturing complexity across intra-block, inter-block, and inter-scale levels. The resulting multi-granularity local entropy patterns (MLEP) are used as input to a standard CNN classifier for distinguishing AI-generated from real images. An overview of the method is shown in Fig. 2, with key components detailed below.

### 3.1   Semantic Suppression via Patch Shuffling

Inspired by previous work [6, 8] that mitigates semantic bias via patch-based processing, we adopt finer patch shuffling to further disrupt image content. Given an input image $X \in \mathbb{R}^{H \times W \times C}$, we first partition it into patches of uniform size of $L \times L$:

$$X = \{X_{i,j} \in \mathbb{R}^{L \times L \times C}\}_{1 \leq i \leq \frac{H}{L}, 1 \leq j \leq \frac{W}{L}}, \tag{1}$$

where $L$ is a small integer (typically $< 8$), and $H, W$ are assumed divisible by $L$. The patches are then randomly permuted, resulting in a visually scrambled image denoted as $\tilde{X}$:

$$\tilde{X} = \{\tilde{X}_{\pi(i,j)} = X_{i,j}\}_{1 \leq i \leq \frac{H}{L}, 1 \leq j \leq \frac{W}{L}}, \tag{2}$$

where $\pi$ is a bijection defining the patch permutation. Note that partitioning and shuffling are applied independently to each color channel.

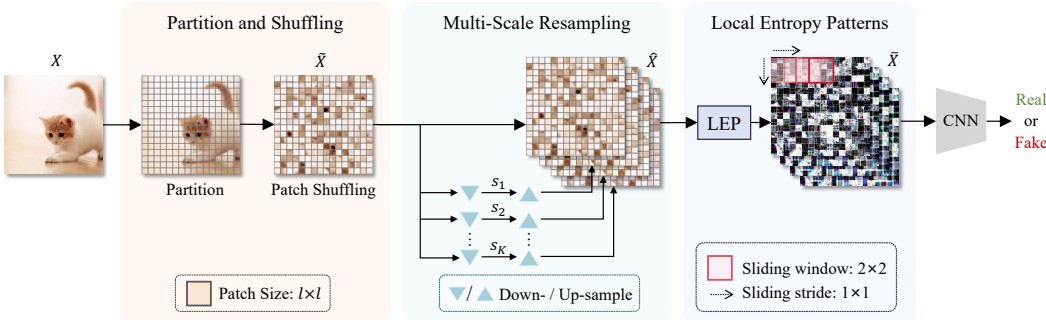

Figure 2: Illustration of AI-generated image detection using multi-granularity local entropy patterns (MLEP), which involves three core steps to obtain the MLEP feature: Patch Shuffling, Multi-Scale Resampling, and Local Entropy Pattern computation. The resulting MLEP features are then fed into a CNN-based classifier (e.g., ResNet) to effectively identify AI-generated images.

## 3.2  Multi-Scale Resampling

Inspired by [7] showing that generative models often use upsampling to produce high-resolution outputs, we propose detecting generation artifacts via multi-scale analysis. We hypothesize that resampling generated images reveals distinctive patterns useful for detection. To this end, we first construct a multi-scale pyramid by resampling the scrambled image $\tilde{X}$ with scale factors $\mathbb{S} = \{s_1, s_2, \ldots, s_K\}$, with each scale $s_k \in (0, 1]$ applied using an interpolation function $\mathrm{Down}(\cdot, s_k)$:

$$\tilde{X}_\vee^{(k)} = \mathrm{Down}(\tilde{X}, s_k), \quad \tilde{X}_\vee^{(k)} \in \mathbb{R}^{\lfloor s_k \cdot H \rfloor \times \lfloor s_k \cdot W \rfloor \times C}, \tag{3}$$

which are then upsampled back to its original shape using an interpolation function $\mathrm{Up}(\cdot, H, W)$:

$$\tilde{X}_\wedge^{(k)} = \mathrm{Up}(\tilde{X}_\vee^{(k)}, H, W), \quad \tilde{X}_\wedge^{(k)} \in \mathbb{R}^{H \times W \times C}. \tag{4}$$

The resulting multi-scale resampling image $\hat{X}$ is created by concatenating all the upsampled images along the channel dimension:

$$\hat{X} = \mathrm{Concat}(\tilde{X}_\wedge^{(1)}, \tilde{X}_\wedge^{(2)}, \ldots, \tilde{X}_\wedge^{(K)}), \quad \hat{X} \in \mathbb{R}^{H \times W \times (C \cdot K)}. \tag{5}$$

## 3.3  Multi-granularity Local Entropy Patterns

The core of our approach is the design of Local Entropy Patterns (LEP), which quantify textural randomness using a $2 \times 2$ sliding window over pixel sets $\hat{X}_{i,j} = \{x_{m,n}\}_{m \in \{i,i+1\}, n \in \{j,j+1\}}$, based on Shannon's definition of information entropy [18]:

$$\mathrm{LEP}\left(\hat{X}_{i,j}\right) = -\sum_{m,n} p(x_{m,n}) \cdot \log_2 p(x_{m,n}), \tag{6}$$

where $p(x_{m,n})$ represents the probability of occurrence of the pixel value $x_{m,n}$ within that specific patch $\hat{X}_{i,j}$. By restricting the sliding window to $2 \times 2$ (four pixels), entropy values are confined to five discrete levels: $\mathbb{V} = \{0, 0.8, 1.0, 1.5, 2\}$, as shown in Fig. 3c. The proof and an efficient computation algorithm for LEP on a $2 \times 2$ window are provided in the supplementary material. With a stride of 1, the $2 \times 2$ sliding window introduces overlap in LEP computation. Due to patch shuffling, this captures both *intra-patch* and *inter-patch* entropy—reflecting local randomness within and across original image regions—as illustrated in Fig. 3b. Applied across multiple scales, LEP further captures *inter-scale* entropy, forming the basis of the final Multi-granularity Local Entropy Patterns (MLEP).

Given the computed MLEP feature maps denoted as $\bar{X} \in \mathbb{V}^{(H-1) \times (W-1) \times (C \cdot K)}$, a representative CNN-based classifier can be trained to differentiate between photographic and AI-generated images. Denoting the classifier as $f$, the training objective is defined using the binary cross-entropy loss:

$$\mathcal{L}_{BCE} = -\frac{1}{N} \sum_{i=1}^{N} \left[ y_i \log(f(\bar{X}_i)) + (1 - y_i) \log(1 - f(\bar{X}_i)) \right], \tag{7}$$

where $y_i$ represents the true labels, $f(\bar{X}_i)$ the predictions, and $N$ the number of training samples.

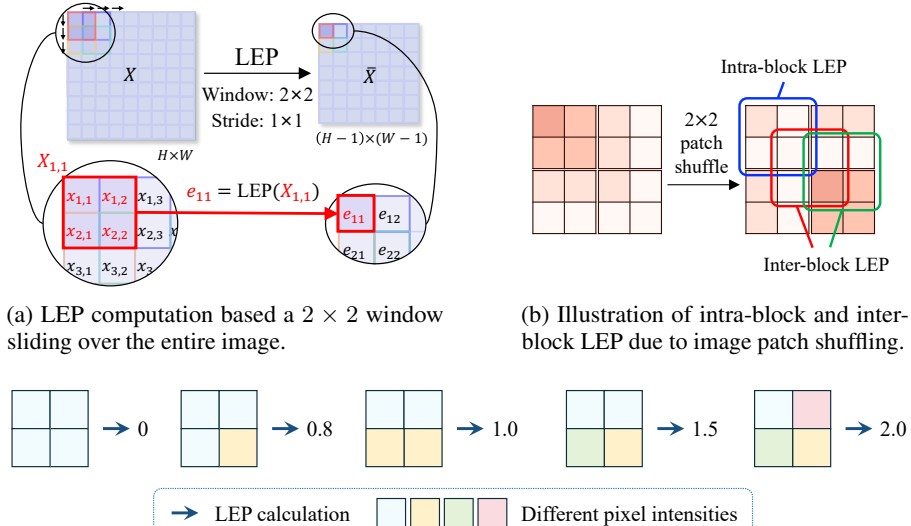

(a) LEP computation based a $2 \times 2$ window sliding over the entire image.

(b) Illustration of intra-block and inter-block LEP due to image patch shuffling.

(c) Five possible LEP values corresponding to different pixel occurrences within a $2\times2$ window.

Figure 3: Illustration of the MLEP computation.

# 4 Experiments

## 4.1 Experimental Settings

**Datasets**    We adopt the cross-dataset setup from [7], using the ForenSynths [5] dataset for training, which includes 20 content categories with 18,000 ProGAN [24] generated images and an equal number of real images from LSUN [25]. Following [7], we train only on four categories: *cars*, *cats*, *chairs*, and *horses*, posing a challenging cross-scene setting. Following [5, 13, 7, 8], we evaluate on synthesized images from 32 image generation models (16 GAN-based and 16 Diffusion-based, including variants). The **GAN-Set** includes ProGAN [24], StyleGAN [26], StyleGAN2 [27], BigGAN [28], CycleGAN [29], StarGAN [30], GauGAN [31], AttGAN [32], BEGAN [33], Cramer-GAN [34], InfoMaxGAN [35], MMDGAN [36], RelGAN [37], S3GAN [38], SNGAN [39], and STGAN [40], with the former seven obtained from the dataset ForenSynths [5] and the latter nine from the dataset GANGen-Detection [41]. The **Diffusion-Set** contains DDPM [2], IDDPM [42], ADM [43], LDM [44], PNDM [45], VQ-Diffusion [46], Stable Diffusion (SD) v1/v2 [44], DALL·E mini [47], three Glide [48] variants[2], and two LDM [44] variants[3]. Of these models, the first eight are sourced from the DiffusionForensics dataset [13], while the remainder are from the UniversalFakeDetect dataset [15]. Furthermore, we include images from two commercial models, Midjourney and DALL·E 2, sourced from the social platform Discord[4] as provided by [7]. Each above AIGI subset comprises an equal number of real samples paired with the corresponding generative counterparts. All test images were obtained according to the instructions provided by [7].

**Implementation details**    All images were resized to $224 \times 224$, with random cropping for training and center cropping for testing. Multiple variants of the patch size ($L$), resampling scales ($\mathbb{S}$), and the classifier backbone were tested with results shown in the ablation study. The training was performed using the Adam optimizer (learning rate of 0.002, batch size of 64). All experiments ran on a server with two NVIDIA RTX A5000 GPUs.

**Baseline methods**    We compare against representative baselines, including CNNDet [5], F3Net [10], LGrad [49], UnivFD [15], CLIPping [16], NPR [7], Zheng [8], FreqNet [21], FatFormer [17],

---

[2]Glide-100-10, Glide-100-27, and Glide-50-27, where Glide-$k$-$l$ means $k$ steps in the first stage and $l$ steps in the second stage of diffusion models.

[3]LDM-200 (LDM with 200 steps) and LDM-200-CFG (LDM with 200 steps with classifier-free diffusion guidance).

[4]https://discord.com/

Table 1: Detection performance in terms of Acc.(%) and A.P.(%) on the GAN-based datasets.

| Method | ProGAN | | StyleGAN | | StyleGAN2 | | BigGAN | | CycleGAN | | StarGAN | | GauGAN | | AttGAN | |
|---|---|---|---|---|---|---|---|---|---|---|---|---|---|---|---|---|
| | Acc. | A.P. | Acc. | A.P. | Acc. | A.P. | Acc. | A.P. | Acc. | A.P. | Acc. | A.P. | Acc. | A.P. | Acc. | A.P. |
| CNNDet [5] | 91.4 | 99.4 | 63.8 | 91.4 | 76.4 | 97.5 | 52.9 | 73.3 | 72.7 | 88.6 | 63.8 | 90.8 | 63.9 | 92.2 | 51.1 | 83.7 |
| F3Net [10] | 99.4 | 100.0 | 92.6 | 99.7 | 88.0 | 99.8 | 65.3 | 69.9 | 76.4 | 84.3 | 100.0 | 100.0 | 58.1 | 56.7 | 85.2 | 94.8 |
| LGrad [49] | 99.0 | 100.0 | 94.8 | 99.9 | 96.0 | 99.9 | 82.9 | 90.7 | 85.3 | 94.0 | 99.6 | 100.0 | 72.4 | 79.3 | 68.6 | 93.8 |
| Ojha [15] | 99.7 | 100.0 | 89.0 | 98.7 | 83.9 | 98.4 | 90.5 | 99.1 | 87.9 | 99.8 | 91.4 | 100.0 | 89.9 | 100.0 | 78.5 | 91.3 |
| Zheng [8] | 99.7 | 100.0 | 90.7 | 95.3 | 97.6 | 99.7 | 67.0 | 67.6 | 85.2 | 92.6 | 98.7 | 100.0 | 57.1 | 56.8 | 79.4 | 87.7 |
| CLIPping [16] | 99.8 | 100.0 | 94.3 | 99.4 | 83.5 | 98.7 | 93.8 | 99.4 | 95.4 | 99.9 | 99.1 | 100.0 | 93.4 | 99.9 | 91.3 | 97.4 |
| NPR [7] | 99.8 | 100.0 | 96.3 | 99.8 | 97.3 | 100.0 | 87.5 | 94.5 | 95.0 | 99.5 | 99.7 | 100.0 | 86.6 | 88.8 | 83.0 | 96.2 |
| FreqNet [21] | 99.6 | 100.0 | 90.2 | 99.7 | 87.9 | 99.5 | 90.5 | 94.6 | 95.8 | 99.6 | 85.6 | 99.8 | 93.4 | 98.6 | 89.8 | 98.8 |
| FatFormer [17] | 99.9 | 100.0 | 97.1 | 99.8 | 98.8 | 99.9 | 99.5 | 100.0 | 99.4 | 100.0 | 99.8 | 100.0 | 99.4 | 100.0 | 99.3 | 100.0 |
| ForgeLens [22] | 99.9 | 100.0 | 90.3 | 98.7 | 94.2 | 98.8 | 98.9 | 99.0 | 99.6 | 99.6 | 99.8 | 100.0 | 99.1 | 99.4 | 90.1 | 90.0 |
| D³ [20] | 99.4 | 100.0 | 94.9 | 99.2 | 95.6 | 99.4 | 99.1 | 100.0 | 92.6 | 98.5 | 95.7 | 99.3 | 97.9 | 99.9 | 84.8 | 92.9 |
| VIBAIGC [23] | 99.9 | 100.0 | 89.0 | 98.5 | 87.0 | 97.2 | 95.3 | 99.1 | 98.7 | 99.7 | 97.7 | 99.9 | 99.3 | 99.9 | 93.4 | 98.1 |
| Ours | 99.6 | 100.0 | 99.6 | 100.0 | 99.9 | 100.0 | 87.1 | 93.6 | 98.3 | 99.3 | 100.0 | 100.0 | 82.0 | 87.9 | 100.0 | 100.0 |

| Method | BEGAN | | CramerGAN | | InfoMaxGAN | | MMDGAN | | RelGAN | | S3GAN | | SNGAN | | STGAN | |
|---|---|---|---|---|---|---|---|---|---|---|---|---|---|---|---|---|
| | Acc. | A.P. | Acc. | A.P. | Acc. | A.P. | Acc. | A.P. | Acc. | A.P. | Acc. | A.P. | Acc. | A.P. | Acc. | A.P. |
| CNNDet [5] | 50.2 | 44.9 | 81.5 | 97.5 | 71.1 | 94.7 | 72.9 | 94.4 | 53.3 | 82.1 | 55.2 | 66.1 | 62.7 | 90.4 | 63.0 | 92.7 |
| F3Net [10] | 87.1 | 97.5 | 89.5 | 99.8 | 67.1 | 83.1 | 73.7 | 99.6 | 98.8 | 100.0 | 65.4 | 70.0 | 51.6 | 93.6 | 60.3 | 99.9 |
| LGrad [49] | 69.9 | 89.2 | 50.3 | 54.0 | 71.1 | 82.0 | 57.5 | 67.3 | 89.1 | 99.1 | 78.5 | 86.0 | 78.0 | 87.4 | 54.8 | 68.0 |
| Ojha [15] | 72.0 | 98.9 | 77.6 | 99.8 | 77.6 | 98.9 | 77.6 | 99.7 | 78.2 | 98.7 | 85.2 | 98.1 | 77.6 | 98.7 | 74.2 | 97.8 |
| Zheng [8] | 67.4 | 98.0 | 74.2 | 93.8 | 71.0 | 93.1 | 68.4 | 89.4 | 98.4 | 99.9 | 70.8 | 69.9 | 72.4 | 94.0 | 92.3 | 100.0 |
| CLIPping [16] | 100.0 | 100.0 | 100.0 | 100.0 | 94.7 | 99.7 | 94.8 | 99.9 | 92.2 | 98.3 | 88.4 | 97.7 | 94.4 | 99.5 | 87.2 | 96.4 |
| NPR [7] | 99.0 | 99.8 | 98.7 | 99.0 | 94.5 | 98.3 | 98.6 | 99.0 | 99.6 | 100.0 | 79.0 | 80.0 | 88.8 | 97.4 | 98.0 | 100.0 |
| FreqNet [21] | 98.8 | 100.0 | 95.1 | 98.2 | 94.5 | 97.3 | 95.1 | 98.2 | 100.0 | 100.0 | 88.4 | 94.3 | 85.3 | 90.5 | 98.8 | 100.0 |
| FatFormer [17] | 99.9 | 100.0 | 98.4 | 100.0 | 98.4 | 100.0 | 98.4 | 100.0 | 99.5 | 100.0 | 99.0 | 100.0 | 98.3 | 99.9 | 98.8 | 99.8 |
| ForgeLens [22] | 88.4 | 97.0 | 87.2 | 93.1 | 87.6 | 92.6 | 87.5 | 92.3 | 92.6 | 93.1 | 98.7 | 99.3 | 86.7 | 91.7 | 90.0 | 95.2 |
| D³ [20] | 89.5 | 97.3 | 95.2 | 99.2 | 95.7 | 99.2 | 94.6 | 99.0 | 91.4 | 97.5 | 98.4 | 99.9 | 93.8 | 98.7 | 93.0 | 98.5 |
| VIBAIGC [23] | 96.7 | 99.5 | 95.3 | 99.0 | 90.5 | 96.5 | 95.3 | 98.7 | 94.7 | 98.7 | 94.6 | 98.9 | 93.4 | 98.2 | 82.4 | 92.5 |
| Ours | 99.4 | 100.0 | 98.5 | 99.8 | 98.0 | 99.8 | 98.9 | 99.8 | 100.0 | 100.0 | 83.4 | 91.7 | 97.6 | 99.7 | 99.9 | 100.0 |

Table 2: Detection performance in terms of Acc.(%) and A.P.(%) on the Diffusion-based datasets.

| Method | ADM | | DDPM | | IDDPM | | LDM | | PNDM | | VQ-Diffusion | | SDv1 | | SDv2 | |
|---|---|---|---|---|---|---|---|---|---|---|---|---|---|---|---|---|
| | Acc. | A.P. | Acc. | A.P. | Acc. | A.P. | Acc. | A.P. | Acc. | A.P. | Acc. | A.P. | Acc. | A.P. | Acc. | A.P. |
| CNNDet [5] | 53.9 | 71.8 | 62.7 | 76.6 | 50.2 | 82.7 | 50.4 | 78.7 | 50.8 | 90.3 | 50.0 | 71.0 | 38.0 | 76.7 | 52.0 | 90.3 |
| F3Net [10] | 80.9 | 96.9 | 84.7 | 99.4 | 74.7 | 98.9 | 100.0 | 100.0 | 72.8 | 99.5 | 100.0 | 100.0 | 73.4 | 97.2 | 99.8 | 100.0 |
| LGrad [49] | 86.4 | 97.5 | 99.9 | 100.0 | 66.1 | 92.8 | 99.7 | 100.0 | 69.5 | 98.5 | 96.2 | 100.0 | 90.4 | 99.4 | 97.1 | 100.0 |
| Ojha [15] | 78.4 | 92.1 | 72.9 | 78.8 | 75.0 | 92.8 | 82.2 | 97.1 | 75.3 | 92.5 | 83.5 | 97.7 | 56.4 | 90.4 | 71.5 | 92.4 |
| Zheng [8] | 72.1 | 78.9 | 78.9 | 80.5 | 49.9 | 52.0 | 99.7 | 100.0 | 90.4 | 96.9 | 99.6 | 100.0 | 94.0 | 99.7 | 87.9 | 96.4 |
| CLIPping [16] | 78.9 | 93.8 | 80.3 | 85.7 | 82.4 | 94.4 | 90.2 | 97.6 | 81.7 | 93.7 | 96.3 | 99.3 | 58.0 | 93.1 | 62.4 | 94.9 |
| NPR [7] | 88.6 | 98.9 | 99.8 | 100.0 | 91.8 | 99.8 | 100.0 | 100.0 | 91.2 | 100.0 | 100.0 | 100.0 | 97.4 | 99.8 | 93.8 | 100.0 |
| FreqNet [21] | 67.2 | 91.3 | 91.5 | 99.8 | 59.0 | 97.3 | 98.9 | 100.0 | 85.2 | 99.8 | 100.0 | 100.0 | 63.9 | 98.1 | 81.8 | 98.4 |
| FatFormer [17] | 70.8 | 93.4 | 67.2 | 72.5 | 69.3 | 94.3 | 97.3 | 100.0 | 99.3 | 100.0 | 99.9 | 100.0 | 61.7 | 96.8 | 84.4 | 98.2 |
| ForgeLens [22] | 69.8 | 92.4 | 52.1 | 52.3 | 62.1 | 75.8 | 99.6 | 100.0 | 83.4 | 97.1 | 99.5 | 100.0 | 93.2 | 99.8 | 63.2 | 87.6 |
| D³ [20] | 89.0 | 97.8 | 85.2 | 94.4 | 87.7 | 96.2 | 88.2 | 96.4 | 90.0 | 96.8 | 96.1 | 99.9 | 98.0 | 99.8 | 93.6 | 98.8 |
| VIBAIGC [23] | 69.3 | 81.8 | 90.2 | 97.7 | 84.6 | 97.1 | 56.8 | 86.6 | 94.8 | 94.3 | 94.2 | 99.5 | 60.0 | 88.7 | 58.3 | 83.2 |
| Ours | 97.0 | 99.8 | 100.0 | 100.0 | 100.0 | 100.0 | 99.8 | 100.0 | 100.0 | 100.0 | 100.0 | 100.0 | 98.5 | 99.9 | 100.0 | 100.0 |

| Method | DALL·E mini | | Glide-100-10 | | Glide-100-27 | | Glide-50-27 | | LDM-200 | | LDM-200-cfg | | Midjourney | | DALL·E 2 | |
|---|---|---|---|---|---|---|---|---|---|---|---|---|---|---|---|---|
| | Acc. | A.P. | Acc. | A.P. | Acc. | A.P. | Acc. | A.P. | Acc. | A.P. | Acc. | A.P. | Acc. | A.P. | Acc. | A.P. |
| CNNDet [5] | 51.8 | 61.3 | 53.3 | 72.9 | 53.0 | 71.3 | 54.2 | 76.0 | 52.0 | 64.5 | 51.6 | 63.1 | 48.6 | 38.5 | 49.3 | 44.7 |
| F3Net [10] | 71.6 | 79.9 | 88.3 | 95.4 | 87.0 | 94.5 | 88.5 | 95.4 | 73.4 | 83.3 | 80.7 | 89.1 | 73.2 | 80.4 | 79.6 | 87.3 |
| LGrad [49] | 88.5 | 97.3 | 89.4 | 94.9 | 87.4 | 93.2 | 90.7 | 95.1 | 94.2 | 99.1 | 95.9 | 99.2 | 68.3 | 76.0 | 75.1 | 80.9 |
| Ojha [15] | 89.5 | 96.8 | 90.1 | 97.0 | 90.7 | 97.2 | 91.1 | 97.4 | 90.2 | 97.1 | 77.3 | 88.6 | 50.0 | 49.8 | 66.3 | 74.6 |
| Zheng [8] | 67.9 | 72.2 | 79.4 | 87.8 | 76.8 | 84.5 | 78.2 | 85.9 | 81.3 | 90.1 | 84.0 | 91.7 | 73.2 | 78.5 | 81.4 | 89.2 |
| CLIPping [16] | 91.1 | 98.6 | 92.0 | 98.6 | 91.2 | 98.8 | 94.3 | 99.3 | 92.8 | 98.9 | 77.4 | 94.3 | 51.1 | 50.7 | 62.6 | 72.3 |
| NPR [7] | 94.5 | 99.5 | 98.2 | 99.8 | 97.8 | 99.8 | 99.1 | 99.9 | 99.1 | 99.9 | 99.0 | 99.8 | 77.4 | 81.9 | 80.7 | 83.0 |
| FreqNet [21] | 97.4 | 99.8 | 88.1 | 96.4 | 84.5 | 96.1 | 86.7 | 96.3 | 97.5 | 99.9 | 97.4 | 99.9 | 55.5 | 65.3 | 52.9 | 61.8 |
| FatFormer [17] | 98.8 | 99.8 | 94.2 | 99.2 | 94.4 | 99.1 | 94.7 | 99.4 | 98.6 | 99.8 | 94.9 | 99.1 | 62.8 | 85.4 | 68.8 | 93.2 |
| ForgeLens [22] | 99.0 | 100.0 | 98.0 | 99.9 | 97.5 | 99.7 | 98.5 | 99.9 | 99.7 | 99.4 | | | 77.6 | 91.7 | 76.8 | 94.9 |
| D³ [20] | 92.8 | 98.3 | 94.4 | 98.8 | 94.7 | 98.8 | 94.9 | 98.9 | 94.8 | 99.4 | 88.3 | 95.9 | 92.5 | 98.6 | 78.0 | 94.6 |
| VIBAIGC [23] | 87.8 | 96.9 | 87.2 | 97.6 | 86.4 | 97.5 | 89.2 | 98.0 | 95.9 | 99.4 | 77.4 | 92.8 | 50.4 | 47.2 | 55.9 | 69.4 |
| Ours | 95.7 | 99.9 | 99.9 | 100.0 | 100.0 | 100.0 | 99.8 | 100.0 | 99.9 | 100.0 | 99.8 | 100.0 | 87.5 | 97.1 | 87.3 | 97.4 |

ForgeLens [22], D³ [20] and VIBAIGC [23]. Accuracy (Acc.) and average precision (A.P.) are used as metrics. Following the same protocol, we re-evaluated CLIPping [16], Zheng [8], FreqNet [21], FatFormer [17], ForgeLens [22], D³ [20], and VIBAIGC [23] using their official open-source implementations, while results for the remaining baselines were taken from [7].

## 4.2 Overall Evaluation of Detection Generalizability

We evaluated the generalization performance of our AIGI detection method across datasets. Accuracy (Acc.) and average precision (A.P.) compared to state-of-the-art GAN- and Diffusion-based methods are reported in Tables 1 and 2, using patch size $L = 2$, resampling scales $\mathbb{S} = \{1, 1/2, 1/4\}$, with a

Table 3: Mean Acc. and A.P. over 16 GAN-based, 16 Diffusion-based, and all 32 datasets.

| Method | GAN-Set | | Diff.-Set | | Mean | |
|---|---|---|---|---|---|---|
| | Acc. | A.P. | Acc. | A.P. | Acc. | A.P. |
| CNNDet [5] | 65.4 | 86.2 | 51.4 | 70.7 | 58.4 | 78.4 |
| F3Net [10] | 78.7 | 90.6 | 83.0 | 93.6 | 80.8 | 92.1 |
| LGrad [49] | 78.0 | 86.9 | 87.2 | 95.2 | 82.6 | 91.1 |
| Ojha [15] | 83.2 | 98.6 | 77.5 | 89.5 | 80.4 | 94.1 |
| Zheng [8] | 80.6 | 89.9 | 80.9 | 86.5 | 80.8 | 88.2 |
| CLIPping [16] | 93.9 | 99.1 | 81.4 | 91.5 | 87.7 | 95.3 |
| NPR [7] | 93.8 | 97.0 | 94.2 | 97.6 | 94.0 | 97.3 |
| FreqNet [21] | 93.1 | 98.2 | 81.7 | 93.8 | 87.4 | 96.0 |
| FatFormer [17] | **99.0** | **100.0** | 84.8 | 95.6 | 91.9 | 97.8 |
| ForgeLens [22] | 93.2 | 96.2 | 85.5 | 93.0 | 89.4 | 94.6 |
| D$^3$ [20] | 94.5 | 98.7 | 91.1 | 97.7 | 92.8 | 98.2 |
| VIBAIGC [23] | 93.5 | 98.1 | 77.2 | 89.8 | 85.4 | 94.0 |
| Ours | 96.4 | 98.2 | **97.8** | **99.6** | **97.1** | **98.9** |

ResNet-50 backbone, which yield the optimal validation results. MLEP consistently achieves top performance across most datasets. Remarkably, it generalizes well to diffusion-generated images, despite being trained solely on GAN-based data (ProGAN [24]). Even on datasets with entirely different content (e.g., face-centric sets like StarGAN, InfoMaxGAN, and AttGAN), MLEP maintains strong performance, underscoring its cross-scene robustness. Table 3 further shows that MLEP outperforms NPR [7], with average gains of 3.1% in Acc. and 1.6% in A.P., despite NPR's already strong results.

## 4.3 Ablation Study

We next conducted a series of ablation studies to evaluate the effectiveness of key components and hyperparameters in the proposed approach.

**Effectiveness of patch shuffling and multi-scale resampling** We first assessed the impact of two key components: patch shuffling and multi-scale resampling. Ablation results in Table 4 show that removing either component noticeably reduces performance, with patch shuffling contributing more. Even without both, LEP alone achieves over 94.3% accuracy, higher than NPR (94.0%) [7], highlighting the effectiveness of entropy-based features.

Table 4: Ablation study on the impact of key components, where PS represents patch shuffling and MR denotes multi-scale resampling.

| LEP | PS | MR | GAN-set | | Diff.-set | | Mean | |
|---|---|---|---|---|---|---|---|---|
| | | | Acc. | A.P. | Acc. | A.P. | Acc. | A.P. |
| ✓ | | | 93.6 | 94.1 | 94.9 | 95.7 | 94.3 | 94.9 |
| ✓ | | ✓ | 93.4 | 94.1 | 95.8 | 96.9 | 94.6 | 95.5 |
| ✓ | ✓ | | 95.7 | 98.2 | 97.5 | 99.6 | 96.6 | 98.9 |
| ✓ | ✓ | ✓ | **96.4** | **98.2** | **97.8** | **99.6** | **97.1** | **98.9** |

**Impact of the resampling interpolation method** We also evaluated the impact of interpolation methods, comparing bilinear, bicubic, and nearest-neighbor (Table 5). Bilinear outperforms nearest-neighbor and performs comparably to bicubic. This might be because bilinear and bicubic blend neighboring pixel values, introducing richer entropy variations, while nearest-neighbor simply copies pixel values, resulting in limited entropy diversity.

Table 5: Impact of the interpolation method.

| Interp. | GAN-set | | Diff.-set | | Mean | |
|---|---|---|---|---|---|---|
| | Acc. | A.P. | Acc. | A.P. | Acc. | A.P. |
| Bilinear | **96.4** | **98.2** | 97.8 | **99.6** | **97.1** | 98.9 |
| Bicubic | 96.2 | 97.9 | **97.9** | 99.3 | 96.9 | **99.1** |
| Nearest | 94.6 | 97.4 | 96.8 | 99.2 | 95.7 | 98.3 |

**Impact of patch size and scale factors** We further examined the effects of patch size and resampling scales by testing different hyperparameter settings, as shown in Table 6. The best performance was achieved with the smallest patch size ($l = 2$), indicating that stronger semantic scrambling improves detection. Moderate multi-scale fusion ($\mathbb{S} = \{1, 1/2, 1/4\}$) also led to optimal results, confirming the benefit of incorporating resampling artifacts.

Table 6: Impact of patch size $L$ and resampling scaling factors $\mathbb{S}$.

| $L$ | $\mathbb{S}$ | GAN-set Acc. | GAN-set A.P. | Diff.-set Acc. | Diff.-set A.P. | Mean Acc. | Mean A.P. |
|---|---|---|---|---|---|---|---|
| | $\{1, 1/2\}$ | 95.8 | 97.8 | 97.5 | 99.6 | 96.6 | 98.7 |
| 2 | $\{1, 1/2, 1/4\}$ | **96.4** | **98.2** | **97.8** | **99.6** | **97.1** | **98.9** |
| | $\{1, 1/2, 1/4, 1/8\}$ | 91.7 | 97.9 | 95.3 | 99.5 | 93.5 | 98.7 |
| | $\{1, 1/2\}$ | 94.5 | 96.6 | 95.5 | 98.8 | 95.0 | 97.7 |
| 4 | $\{1, 1/2, 1/4\}$ | 94.5 | 96.8 | 96.6 | 99.1 | 95.5 | 97.9 |
| | $\{1, 1/2, 1/4, 1/8\}$ | 94.2 | 96.4 | 96.5 | 98.8 | 95.4 | 97.6 |
| | $\{1, 1/2\}$ | 93.9 | 95.8 | 95.4 | 97.7 | 94.7 | 96.7 |
| 8 | $\{1, 1/2, 1/4\}$ | 94.0 | 96.5 | 95.8 | 99.1 | 94.9 | 97.8 |
| | $\{1, 1/2, 1/4, 1/8\}$ | 94.4 | 96.0 | 95.8 | 98.1 | 95.1 | 97.0 |

**Impact of sliding window stride** We evaluated the effect of stride in the $2 \times 2$ sliding window for LEP computation, testing strides of 1 and 2 (Table 7). A stride of 1 significantly outperforms 2, highlighting the importance of inter-block entropy in MLEP. This supports our multi-granularity design, which captures both intra- and inter-block texture patterns as depicted in Fig. 3.

Table 7: Impact of sliding window stride.

| Stride | GAN-set Acc. | GAN-set A.P. | Diff.-set Acc. | Diff.-set A.P. | Mean Acc. | Mean A.P. |
|---|---|---|---|---|---|---|
| 1 | **96.4** | **98.2** | **97.8** | **99.6** | **97.1** | **98.9** |
| 2 | 94.9 | 97.3 | 97.0 | 99.5 | 95.9 | 98.4 |

**Compatibility with various backbones** Lastly, we assessed the compatibility of MLEP with various ResNet backbones [50], including ResNet-18, 34, 50, and 101. As shown in Table 8, all variants achieved strong performance, with slight gains from larger models. This confirms the generality and scalability of the proposed feature extraction method.

Table 8: Evaluation on various ResNets.

| Backbone | GAN-set Acc. | GAN-set A.P. | Diff.-set Acc. | Diff.-set A.P. | Mean Acc. | Mean A.P. |
|---|---|---|---|---|---|---|
| ResNet-18 | 96.0 | 97.9 | 97.7 | 99.5 | 96.8 | 98.7 |
| ResNet-34 | 96.1 | 98.2 | 97.7 | **99.7** | 96.9 | 98.9 |
| ResNet-50 | 96.4 | 98.2 | 97.8 | 99.6 | 97.1 | 98.9 |
| ResNet-101 | **96.4** | **98.3** | **97.8** | 99.6 | **97.1** | **99.0** |

**Influence of generation model's text prompts** In text-to-image diffusion models, the specificity of input prompts may greatly affect the visual quality and details of generated images. To inspect the influence of text prompts on AIGI detection performance, we conducted an additional experiment using DiffusionDB[51], a large dataset with 14 million Stable Diffusion images generated from 1.8 million unique prompts. We randomly selected two subsets of 3,000 images, one set generated from complex prompts (over 200 characters with keywords like "high quality," "detailed," and "realistic") and the other from simple prompts (under 100 characters and without those keywords). We evaluated our trained detector on both subsets and found almost no difference in detection accuracy: 99.65% accuracy on the simple set and 99.62% accuracy on the complex set. This further demonstrates the generalizability of the proposed method over different types of AI-generated content.

## 4.4 Interpretability of MLEP

To illustrate the effectiveness of MLEP for AIGI detection, we conducted a set of qualitative analysis detailed as follows.

**Entropy patterns between real and AI-generated images** We first visualize LEP maps for several real–fake image pairs, along with their differences in the pixel, entropy, and Fourier domains. Here, "fake" refers to AI-reconstructed images resembling the originals. Since LEP values are sparse and capped at 2, we normalize them to $[0, 255]$ for visualization. As shown in Fig. 4, LEP differences are

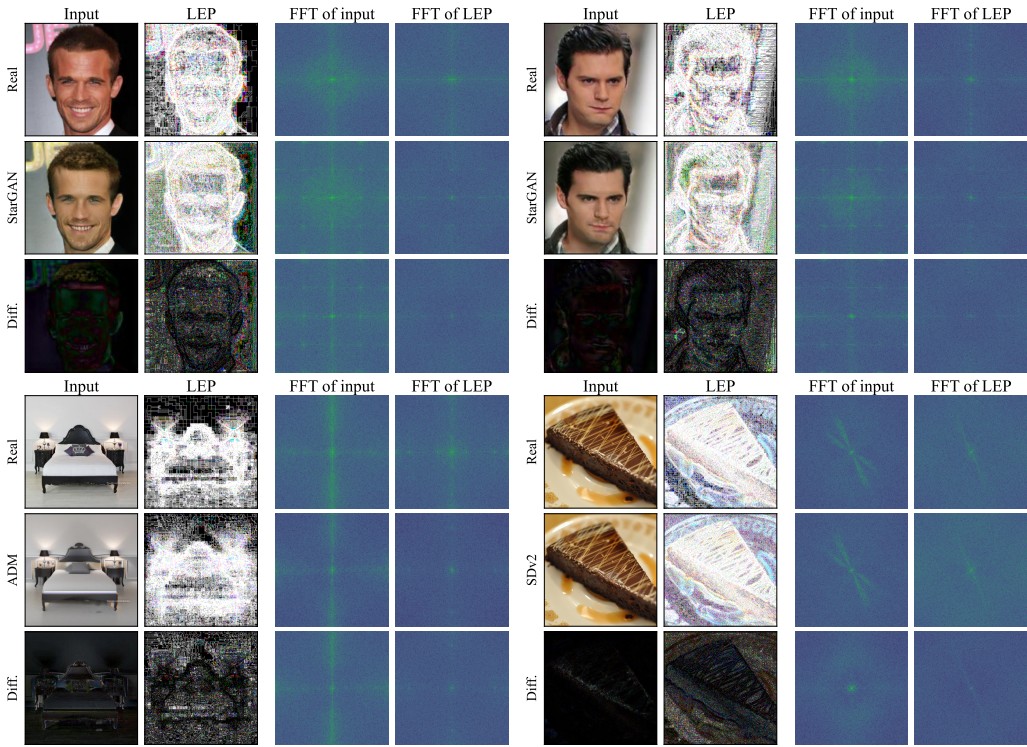

Figure 4: Visualization of local entropy patterns for several real–fake image pairs, along with their differences in the pixel, entropy, and Fourier domains.

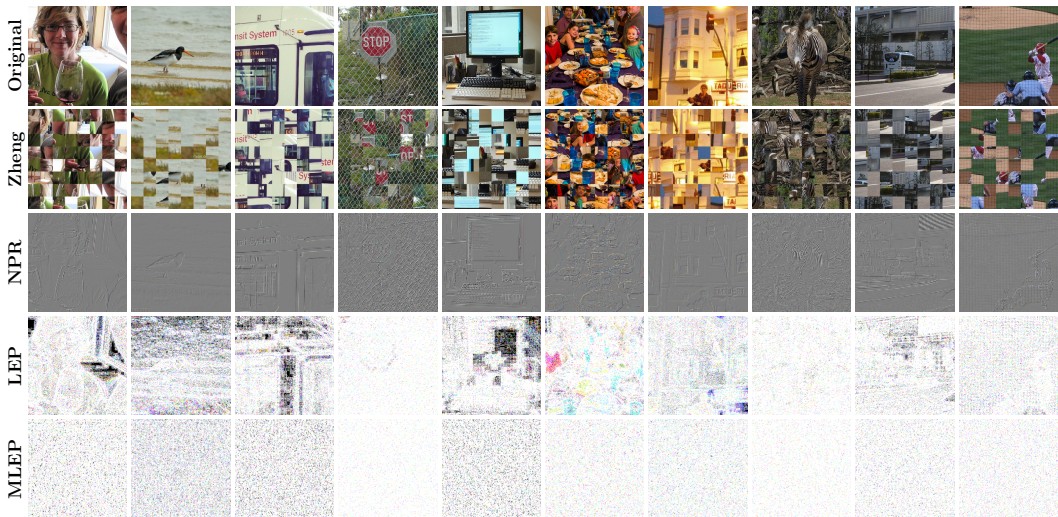

Figure 5: Qualitative comparison among Zheng [8], NPR [7], and our method. LEP preserves minimal visible semantics, while MLEP (without resampling) further suppresses semantic content.

far more pronounced than pixel-level differences, especially for high-quality generations like Stable Diffusion v2, where pixel differences are visually negligible. In the frequency domain, real–fake differences show more consistent patterns than in the pixel space, supporting content-agnostic detection. These results highlight LEP's ability to amplify real–fake discrepancies while minimizing semantic interference.

**Semantic suppression capability of MLEP** We further examine the semantic suppression capability of MLEP compared to two competitive methods: Zheng [8] and NPR [7]. Fig.5 visualizes feature maps of LEP and MLEP (without multi-scale resampling), alongside those from Zheng [8] and NPR [7]. The $32 \times 32$ shuffled patches in Zheng[8] still retain noticeable semantic cues both locally and globally. NPR [7] produces edge-like features by computing pixel differences, leaving much of the original semantics intact. In contrast, LEP substantially suppresses semantic content by highlighting pixel-level randomness, and MLEP further eliminates it through fine-grained patch shuffling, enabling learning content-agnostic representation for AIGI detection.

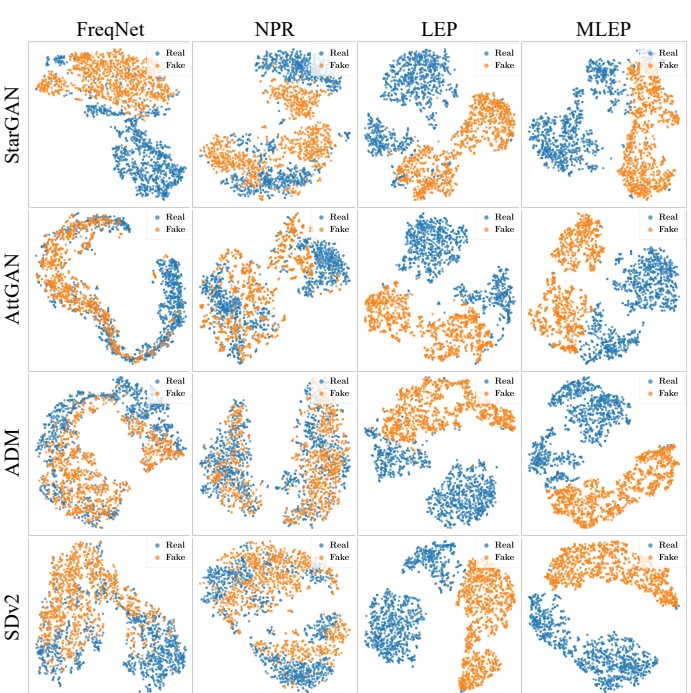

Figure 6: t-SNE visualization of real vs. fake samples.

**Feature distribution of real and AI-generated images** Finally, Fig.6 visualizes the t-SNE distribution [52] of real and fake samples based on the final feature layer of a ResNet-50 classifier, comparing our method with two competitive baselines—NPR [7] and FreqNet [21]—which also use ResNet-50. We showcase results on four generative models: StarGAN [30], AttGAN [32], ADM [43], and SDv2 [44]. The proposed local entropy patterns (LEP) achieve noticeably cleaner real–fake separation than the baselines, and MLEP further enhances this distinction, demonstrating stronger discriminative capability for AIGI detection.

## 5   Conclusion and Limitations

This paper explores the use of entropy as a cue for detecting AI-generated images (AIGI) and introduces Multi-granularity Local Entropy Patterns (MLEP), a set of entropy-based feature maps derived from shuffled small patches across multiple image scales. MLEP captures pixel relationships across spatial and scale dimensions while disrupting image semantics, thereby mitigating content bias. Using MLEP as input, a CNN-based classifier (e.g., ResNet) achieves robust and highly generalizable detection performance.

**Limitations** Nonetheless, limitations still remain. The paper does not explicitly address the robustness of the detector under common image post-processing operations. In fact, when applying different levels of JPEG compression, blurring, or noise, we observe a 17% to 45% drop in detection accuracy—performance that is less satisfactory compared to methods explicitly optimized for robustness. This limitation stems from the fact that MLEP was not specifically designed to handle such transformations, and no special data augmentation techniques were employed during training. Instead, the paper is focused on exploring the potential of using information entropy as a discriminative signal for AIGI detection and on revealing the intrinsic differences in local entropy patterns between real and AI-generated images. Interestingly, the proposed method shows strong robustness to image rescaling: when images are downsampled to half their original resolution, the mean detection accuracy remains above 92.4% (only 4.7% drop). We attribute this to the multi-scale resampling strategy used during training, which effectively introduced resolution variability as a form of implicit data augmentation. Moreover, entropy is computed only within small $2 \times 2$ windows, as using larger windows would exponentially increase computational complexity. In the future, more efforts could be devoted to improve the robustness and computation efficiency of entropy-based approach.

## Acknowledgements

This work is supported by the National Natural Science Foundation of China under grants 62201107, U22A2096, 62502060, 62402073, and 62221005, in part by the Natural Science Foundation of Chongqing under grant CSTB2023NSCQ-LZX0061, in part by the Science and Technology Innovation Key R&D Program of Chongqing under grant CSTB2023TIAD-STX0016, and in part by the Science and Technology Research Program of Chongqing Municipal Education Commission under grants KJQN202300606, KJQN202300619, and KJQN202500649. Special thanks are extended to Prof. Nannan Wang, Prof. Xiuli Bi, Prof. Gwanggil Jeon, and Prof. Touradj Ebrahimi for their invaluable guidance, insightful feedback, and continuous encouragement throughout this research.

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
