# OpenReview forum: "MLEP: Multi-granularity Local Entropy Patterns for Generalized AI-generated Image Detection"
_NeurIPS.cc/2025/Conference — NeurIPS 2025 poster_

### Official Review · Reviewer_YpTU · 2025-07-02

**Clarity:** 3
**Significance:** 3
**Originality:** 3
**Rating:** 5
**Confidence:** 4

**Summary:**

This paper introduces MLEP (Multi-granularity Local Entropy Patterns), a novel method for detecting AI-generated images by analyzing local image entropy. Instead of relying on semantic or pixel-level artifacts, MLEP computes entropy features from shuffled small patches across multiple scales, effectively capturing pixel relationships while disrupting image semantics to reduce content bias. A CNN is then trained on these features for robust classification. Experiments across 32 generative models show that MLEP significantly improves both accuracy and generalization, outperforming prior state-of-the-art methods in open-world detection scenarios.

**Questions:**

Would you consider evaluating your method on the Synthbuster benchmark, particularly those based on the RAISE-1k dataset? This benchmark includes images generated by some of the most recent and challenging diffusion models, and testing on it—even at a small scale—would provide valuable insights into the practical robustness and generalization ability of your method.

**Ethical Concerns:**

["NO or VERY MINOR ethics concerns only"]

**Limitations:**

No negative societal impact.

**Paper Formatting Concerns:**

No formatting issues.

**Quality:**

3

**Strengths And Weaknesses:**

Strengths:
1. The paper is the first to propose using image entropy as a core feature for AI-generated image detection. This is a fresh perspective compared to commonly used spatial or frequency cues.
2. The motivation is clearly explained, building on limitations of previous works (e.g., semantic bias in pixel-based methods) and supporting the idea with preliminary entropy analysis. The Multi-granularity Local Entropy Patterns (MLEP) is a thoughtfully designed contribution that balances semantic disruption and fine-grained pixel relationship capture.
3. Extensive experiments across 32 generative models in open-world settings demonstrate significant improvements in both accuracy and generalization over state-of-the-art training-free and training-based detectors.
4. The paper is well-written and easy to follow, with a logical flow from motivation to methodology, experiments, and analysis. Visuals and tables support the claims effectively.
Weaknesses:
Although the paper shows strong generalization across many generative models, it lacks evaluation on more recent and challenging generators, such as SD3, Flux, etc. I think the authors should test and compare their performance on Synthbuster benchmark (based on RAISE-1k). Including even a small-scale evaluation on these newer sources would provide stronger evidence of real-world applicability and robustness against the latest generation techniques.

---

> ### Author Rebuttal · Authors · 2025-07-29
>
> Thank you for your positive feedback. In response to your concern about the generalization ability of the proposed approach against more recent generators, we conducted additional experiments on several other released generative image datasets, including high-quality images produced by GPT-4o, Flux, and Stable Diffusion. Specifically, we evaluated our model—originally trained on ProGAN—against these datasets in a fully out-of-distribution (i.e., cross-generator) setting. The results, summarized in the table below, report mean detection accuracy (Acc.) and average precision (A.P.), with balanced numbers of real and fake samples in each case. As shown, our method consistently delivers strong performance across most tested datasets, demonstrating its robust generalization ability in detecting AI-generated content from previously unseen generators.
>
> | Dataset | MLEP (Ours) | NPR (CVPR 2024) |
> |:------:|:------:|:------:|
> | GPT-4o | **100.0% / 100.0** | 100.0% / 100.0% |
> | Flux | 84.8% / 96.7% | **94.2% / 98.9%** |
> | firefly (from Synthbuster) | 93.3% / 97.9% | **93.4% / 99.1%** |
> | sdxl (from Synthbuster) | **99.9% / 100.0%** | 99.9% / 100.0% |
> | Midjourney-v5 (from Synthbuster) | 98.7% / **100.0%** | **99.4%** / 99.8% |
> | SD v1.4 (from GenImage) | **98.6% / 99.9%** | 92.3% / 95.6% |
> | SD v1.5 (from GenImage) | **98.7% / 99.8%** | 92.0% / 95.5% |
> | Wukong (from GenImage) | **98.6% / 99.9%** | 87.9% / 92.8% |
> | **Mean** | **96.6% / 99.3%** | 94.9% / 97.7% |
>
> > The two values in each cell of the above table denote Acc./A.P. respectively.

---

> ### Comment · Reviewer_YpTU · 2025-08-06
>
> The rebuttal has addressed all the concerns and I will keep the score to accept the paper.

---

> > ### Author Response · Authors · 2025-08-07
> >
> > Thank you very much for your valuable efforts in reviewing the paper and for your continued support.

---

### Official Review · Reviewer_EVyz · 2025-07-02

**Clarity:** 2
**Significance:** 3
**Originality:** 3
**Rating:** 4
**Confidence:** 4

**Summary:**

This paper proposes MLEP for detecting AI-generated images (AIGIs).
MLEP extracts local pixel randomness as an entropy-based feature, enabling semantic-agnostic, multigranularity discrimination between generated and real images without relying on high-level semantic cues.

To achieve this, the method employs:

Small patch shuffling to eliminate semantic clues,

Multi-scale resampling to expose upsampling/downsampling artifacts, and

A 2×2 sliding window to compute local entropy, capturing the statistical relationships between neighboring pixels.

The extracted MLEP features are fed into a CNN classifier. Experiments show that this approach achieves higher detection accuracy than existing methods across various image generation models.

**Questions:**

If the following four issues are addressed, the paper's score could be improved:
In addition to Weaknesses 1, 2, 3 and 4,
5. Another entropy-based approach for AIGI detection:
Reference [2] also uses entropy-based features for detecting AI-generated images, but it appears to be missing from the paper's discussion.
It would be helpful if the authors could briefly explain the additional contributions or key differences of the proposed MLEP compared to [2].
[2] Cozzolino, Davide, et al. "Zero-shot detection of AI-generated images." European Conference on Computer Vision. Cham: Springer Nature Switzerland, 2024.

**Ethical Concerns:**

["NO or VERY MINOR ethics concerns only"]

**Final Justification:**

The authors have addressed my concerns and weaknesses, particularly W1, W2, W4, and Q5, with additional experiments and well-reasoned responses. These clarifications resolved my doubts.
Regarding W3, while the authors acknowledged the limitations I pointed out, they also provided insightful intuitions that helped reinforce the credibility of their approach.

**Limitations:**

yes

**Quality:**

2

**Strengths And Weaknesses:**

Strengths

The observation that the 2×2 window-based entropy values converge into five discrete levels serves as a concise yet original design insight. This differentiates the method from existing approaches such as FFT-based analysis and CLIP-based models.

The proposed method demonstrates superior performance compared to other baselines on standard benchmarks.

Weaknesses

Lack of details on image generation prompts and its implications for MLEP behavior:
The paper provides insufficient information about the conditions under which the images were generated (e.g., specific prompts used). In text-to-image diffusion models, not only the content but also the visual detail level varies significantly with prompt specificity (e.g., "high-quality," "realistic," camera model descriptions, etc.).
When simple category names are used as prompts, diffusion guidance often leads to smoothed-out backgrounds, which may be easily detected by MLEP's ResNet module due to low local entropy. This raises concerns about scene generalization, and further discussion or additional experiments on the impact of prompt specificity are needed.

Potential real/fake performance imbalance:
The paper should analyze the detection performance for real and fake images separately. Relying solely on AP (Average Precision) may underestimate imbalances between real and fake detection accuracy. A confusion matrix, as well as metrics like FPR (False Positive Rate) and FNR (False Negative Rate), would help clarify whether misclassifications are evenly distributed or skewed toward either class.

Lack of robustness tests against image post-processing:
Although robustness is briefly mentioned in the conclusion, experiments involving common post-processing operations such as JPEG compression or noise addition are necessary—especially since these are standard in real-world scenarios and have been evaluated in many AIGI detection studies. The inclusion of such experiments would strengthen the practical relevance of the method.

Limited comparison and outdated baselines:
While ForenSynths is used for consistency with prior work, concerns have been raised about the visual quality of its generated images [1]. Additionally, some baselines are outdated, which casts doubt on the rigor of the performance comparison. Including more recent methods would make the evaluation more convincing.

[1] Gye, Seoyeon, et al. "Reducing the Content Bias for AI-generated Image Detection." 2025 IEEE/CVF Winter Conference on Applications of Computer Vision (WACV). IEEE, 2025.

---

> ### Author Rebuttal · Authors · 2025-07-29
>
> ## Response to Weakness 1 - Impact of prompts of text-to-image generation models
> Thank you for your thoughtful comment. We agree that in text-to-image diffusion models, prompt specificity can greatly affect the visual quality and details of generated images. We acknowledge that our manuscript did not provide enough details about the prompts, as the test images were sourced from public datasets that cover diverse content but do not specify the prompt details for each image.
>
> To address this, we conducted an additional experiment using DiffusionDB (ACL 2023), a large dataset with 14 million Stable Diffusion images generated from 1.8 million unique prompts. We randomly selected two subsets of 3,000 images each: one from complex prompts (over 200 characters with keywords like “high quality,” “detailed,” and “realistic”) and one from simple prompts (under 100 characters and without those keywords). We evaluated our trained detector on both subsets and found almost no difference in performance (see Table below). This shows that MLEP remains effective even with high-detail prompts, demonstrating its robustness to variations in prompt complexity and image detail.
> | Text Prompt | Acc. (fake only) |
> |------|------|
> | Simple  | 99.65% |
> | Detailed | 99.62% |
>
>
>
> ## Response to Weakness 2 - Potential real/fake performance imbalance
> First, we would like to clarify that all of our evaluations were conducted under balanced settings—each set of AI-generated images was paired with a comparable number of real image samples. Therefore, we opted to report standard metrics such as accuracy and average precision, which are commonly used by baseline methods, including NPR. In response to your suggestion, we conducted additional analyses to examine detection performance for real and fake images separately, and further included false positive rate (FPR) and false negative rate (FNR). The results indicate that misclassifications are generally well-balanced, with the FPR being slightly higher than the FNR, suggesting that the model maintains relatively even sensitivity across both classes.
> | Dataset | Acc. | A.P. | Acc.-Real | Acc.-Fake | FNR | FPR |
> |------|------|------|------|------|------|------|
> | GAN-sets | 98.2% | 96.4% | 94.3% | 98.4% | 1.57% | 5.72% |
> | DM-sets | 99.6% | 97.9% | 96.9% | 98.8% | 1.2% | 3.1% |
> | Mean | 98.9% | 97.1% | 95.6% | 98.6% | 1.39% | 4.41% |
>
>
> ## Response to Weakness 3 - Lack of robustness tests against image post-processing
> Thank you for raising this important point. We acknowledge that the current manuscript does not explicitly address the robustness of our method under common image post-processing operations. Indeed, robustness is not the primary strength of the proposed MLEP approach. For example, when applying different levels of JPEG compression, blurring, or noise, we observed a 17% to 45% drop in detection accuracy—performance that is less satisfactory compared to methods explicitly optimized for robustness. This limitation stems from the fact that MLEP was not specifically designed to handle such transformations, and no data augmentation techniques were employed during training. Instead, our focus was on exploring the potential of information entropy as a discriminative signal for AIGI detection and on revealing the intrinsic differences in local entropy patterns between real and AI-generated images. Interestingly, our method shows strong robustness to image rescaling: when images are downsampled to half their original resolution, the mean detection accuracy remains above 92.4% (only 4.7% drop). We attribute this to the multi-scale resampling strategy used during training, which effectively introduced resolution variability as a form of implicit data augmentation.
>
> We fully agree that robustness to post-processing is critical for practical deployment. Promising directions for future work include incorporating a broader range of image transformations during training and developing more advanced entropy-based features that are resilient to such distortions. We plan to explore these aspects in greater depth in our subsequent research. Thank you for your understanding.
>
>
>
> ## Response to Weakness 4 - Limited comparison and outdated baselines
> Thank you for your comments. To address your concern, we conducted additional experiments on several other released generative image datasets, including high-quality images produced by GPT-4o and Stable Diffusion. Specifically, we evaluated our model—originally trained on ProGAN—against these datasets in a fully out-of-distribution (i.e., cross-generator) setting. The results, summarized in the table below, report mean detection accuracy (Acc.) and average precision (AP), with balanced numbers of real and fake samples in each case. As shown, our method consistently delivers strong performance across most tested datasets, demonstrating its robust generalization ability in detecting AI-generated content from previously unseen generators.
>
> | Dataset | MLEP (Ours) | NPR (CVPR 2024) |
> |:------:|:------:|:------:|
> | GPT-4o | **100.0% / 100.0** | 100.0% / 100.0% |
> | Flux | 84.8% / 96.7% | **94.2% / 98.9%** |
> | firefly (from Synthbuster) | 93.3% / 97.9% | **93.4% / 99.1%** |
> | sdxl (from Synthbuster) | **99.9% / 100.0%** | 99.9% / 100.0% |
> | Midjourney-v5 (from Synthbuster) | 98.7% / **100.0%** | **99.4%** / 99.8% |
> | SD v1.4 (from GenImage) | **98.6% / 99.9%** | 92.3% / 95.6% |
> | SD v1.5 (from GenImage) | **98.7% / 99.8%** | 92.0% / 95.5% |
> | Wukong (from GenImage) | **98.6% / 99.9%** | 87.9% / 92.8% |
> | **Mean** | **96.6% / 99.3%** | 94.9% / 97.7% |
>
> > The two values in each cell of the above table denote Acc./A.P. respectively.
>
> Moreover, in response to your suggestion, we compared our method with several competitive approaches recently published in 2025. Using their publicly available pretrained models, we evaluated all methods—including ours—on the same set of GAN- and diffusion-based datasets. The comparison results, reported in terms of accuracy (Acc.) and average precision (AP), are summarized in the table below (with balanced real and fake samples). As shown, MLEP consistently outperforms these state-of-the-art methods, demonstrating its effectiveness and robustness. We hope these results offer a more comprehensive and convincing evaluation of our approach.
> | Baseline |  Venue | GAN-sets |    DM-sets |    Mean |
> |:------:|:------:|:------:|:------:|:------:|
> | D3 [1] | CVPR 2025 | 94.5% / **98.7%** | 91.1% / 97.7% | 92.8% / 98.2% |
> | VIBAIGC [2] | CVPR 2025 | 93.2% / 98.6% | 77.0% / 88.1% | 85.1% / 93.3% |
> | ForgeLens [3] | ICCV 2025 | 93.2% / 96.2% | 85.5% / 93.0% | 89.4% / 94.6% |
> | **MLEP** | Ours | **96.4%** / 98.2% | **97.8%** / **99.6%** | **97.1%** / **98.9%**|
>
> > Two values in the above table are Acc./A.P. respectively. References:
>
>     [1] D3: Scaling Up Deepfake Detection by Learning from Discrepancy, CVPR 2025
>     [2] Towards Universal AI-Generated Image Detection by Variational Information Bottleneck Network, CVPR 2025
>     [3] ForgeLens: Data-Efficient Forgery Focus for Generalizable Forgery Image Detection, ICCV 2025
>
>
>
>
> ## Response to Question 5 - Differences between MLEP and ZED
> The key difference between our method MLEP and ZED (Zero-Shot Detection of AI-Generated Images) lies in how they leverage entropy for detecting AI-generated images. The specific differences are given below:
>
> > MLEP detects AI-generated images by explicitly computing Shannon entropy over small pixel patches (typically 2×2) to capture local statistical irregularities. It treats entropy as a spatial signal and uses patch scrambling to eliminate semantic bias. A CNN is then trained on both real and synthetic images to learn discriminative entropy patterns across multiple scales. The core insight is that real images contain more structured high-frequency details, reflected in distinct entropy distributions.
>
> > In contrast, ZED is a zero-shot method trained only on real images. It uses a pretrained lossless encoder to estimate pixel-level probabilities and measures the difference between model-based entropy and negative log-likelihood (NLL)—known as the coding cost gap—as the detection signal. This gap highlights inconsistencies commonly found in AI-generated images.
>
> It is hard to say MLEP is much more advanced, but the additional contributions of MLEP compared to ZED include:
>
> - Explicit entropy: Computes Shannon entropy over small patches without relying on learned distributions.
> - Multi-scale analysis: Captures entropy patterns across resolutions for better robustness.
> - Semantic-agnostic: Uses patch scrambling to avoid reliance on content semantics.
> - Efficient and lightweight: Uses vectorized ops and standard CNNs for fast, scalable detection.
> - Trainable end-to-end: Optimized directly for detection, unlike ZED’s fixed encoder.
>
> Nevertheless, both methods tackle AIGI detection using entropy from different angles, offering complementary insights and demonstrating entropy’s potential as a strong cue for identifying AI-generated images.

---

> > ### Comment · Reviewer_EVyz · 2025-08-07
> >
> > I appreciate the authors’ thoughtful and thorough rebuttal. The concerns I raised regarding W1, W2, W4, and Q5 were well addressed with additional experiments and clear explanations, which resolved my doubts. For W3, while the authors acknowledged the limitation I pointed out, they provided helpful intuitions that reinforced the credibility of their method. I believe the experimental results presented in the rebuttal, especially those related to W1 and W4, are valuable and should be included in the main paper to enhance its impact. Based on the improved understanding, I am raising my score to 4.

---

> > > ### Author Response · Authors · 2025-08-08
> > >
> > > Thank you for your valuable review and continued support. We will include those updated results in the final manuscript if the paper luckily gets accepted.

---

### Official Review · Reviewer_3jko · 2025-07-02

**Clarity:** 3
**Significance:** 3
**Originality:** 3
**Rating:** 4
**Confidence:** 1

**Summary:**

This paper introduces a method for detecting AI-generated images called Multi-granularity Local Entropy Patterns (MLEP). The core idea is to leverage image entropy as a distinguishing feature.

The method first shuffles small image patches to suppress semantic content, then creates a multi-scale image pyramid through resampling. Finally, it computes local entropy maps across this pyramid, capturing pixel relationships at multiple granularities. These MLEP feature maps are fed into a standard CNN classifier.

The approach demonstrates significant improvements in both accuracy and generalization for AIGI detection across a wide range of 32 different generative models, outperforming state-of-the-art methods.

**Questions:**

1. Can the method generalize to recent state-of-the-art ImageNet-scale image generation models such as GPT-4o, Flux, or Imagen 3?

2. Can the method be extended to handle AI-generated video content as well?


For AC：
Please note that I’m not very familiar with this specific area, so my comments are offered with that limitation in mind.

**Ethical Concerns:**

["NO or VERY MINOR ethics concerns only"]

**Final Justification:**

Thanks for the rebuttal; it effectively solved my question， and I keep my initial positve score

**Limitations:**

yes

**Quality:**

3

**Strengths And Weaknesses:**

Strengths：
1  The use of local entropy as the primary cue for AIGI detection is novel and well-motivated. The initial analysis in the introduction provides a clear intuition that real images tend to have higher local entropy.

2. The proposed pipeline, combining fine-grained patch shuffling with multi-scale analysis, is a reasonable approach to suppressing semantic content while amplifying subtle generative artifacts.

3. The method demonstrates excellent generalization capabilities, achieving state-of-the-art performance across different generative models.

Weaknesses:
I believe the method relies heavily on strong priors, which leads to a lack of high-frequency details in the generated images. I'm interested in understanding how this method performs when applied to recent state-of-the-art high-resolution image generation models, such as GPT-4o and other methods that can produce images at 2K or 4K resolution.

---

> ### Author Rebuttal · Authors · 2025-07-29
>
> Thank you for your valuable comments. Indeed, our work is based on a strong assumption that AI-generated images (AIGIs) may exhibit different levels of high-frequency components or randomness compared to real photographic images. This stems from the observation that generative models primarily focus on producing semantically coherent content, but often fail to replicate the low-level noise characteristics inherent in real imaging processes. This motivated our curiosity to investigate the differences between real and AI-generated images through the lens of information entropy—a widely used metric for quantifying image randomness. To this end, we conducted a preliminary study comparing the local entropy distributions (based on 2×2 pixel patches) of real and AI-generated images. As shown in Fig. 1 of the submitted manuscript, real images consistently display a higher likelihood of containing blocks with an entropy value of 2.0, indicating the presence of finer local details. This finding supports our hypothesis that AIGIs, while semantically plausible, tend to exhibit greater structural randomness and lack the detailed local variability seen in real images. Building on this insight, we proposed MLEP (Multi-granularity Local Entropy Patterns), a method that leverages local entropy as a discriminative signal for AIGI detection. The core idea of MLEP is to suppress high-level semantic cues by scrambling small image patches, thereby forcing the detector to rely on low-level entropy patterns rather than content semantics, which may introduce bias. By analyzing the distribution of information randomness both within local patches and across randomly sampled regions, MLEP operates in a content-agnostic manner. Furthermore, we introduce a multi-scale resampling strategy to generalize the entropy estimation across different resolutions, enhancing the method’s robustness in diverse scenarios.
>
> In response to your concerns and two questions about the generalization capability of the proposed method on newly emerging, higher-quality generative images, we conducted additional experiments using the suggested datasets, including images generated by GPT-4o, Flux, Midjourney V5, and Stable Diffusion. Specifically, we evaluated the model originally trained on ProGAN against these newly introduced datasets. Each set of generated images were paired with comparable number of real images selected at random. These experiments were thus conducted entirely in out-of-distribution (i.e., cross-generator) settings. The mean detection accuracy (Acc.) and average precision (A.P.) are summarized in the table below. The SOTA method, i.e. NPR (CVPR 2024), are included for comparison as a reference. As shown, our method maintains consistently high performance across all these novel datasets, demonstrating its strong generalization ability in out-of-distribution scenarios.
>
> | Dataset | MLEP (Ours) | NPR (CVPR 2024) |
> |:------:|:------:|:------:|
> | GPT-4o | **100.0% / 100.0** | 100.0% / 100.0% |
> | Flux | 84.8% / 96.7% | **94.2% / 98.9%** |
> | firefly (from Synthbuster) | 93.3% / 97.9% | **93.4% / 99.1%** |
> | sdxl (from Synthbuster) | **99.9% / 100.0%** | 99.9% / 100.0% |
> | Midjourney-v5 (from Synthbuster) | 98.7% / **100.0%** | **99.4%** / 99.8% |
> | SD v1.4 (from GenImage) | **98.6% / 99.9%** | 92.3% / 95.6% |
> | SD v1.5 (from GenImage) | **98.7% / 99.8%** | 92.0% / 95.5% |
> | Wukong (from GenImage) | **98.6% / 99.9%** | 87.9% / 92.8% |
> | **Mean** | **96.6% / 99.3%** | 94.9% / 97.7% |
>
> > The two values in each cell of the above table denote Acc./A.P. respectively.
>
> Unfortunately, our method is less effective on generated video frames, with detection accuracy dropping by more than 30% in most cases. This is primarily because MLEP was not designed to handle image or video compression, and most available video clips have undergone some level of compression. In contrast, our study focused on exploring information entropy as a discriminative signal for AIGI detection, highlighting the intrinsic differences in local entropy patterns between real and AI-generated images, while placing less emphasis on robustness optimization. We fully agree that a robust image-based detector should generalize well to AI-generated video content. However, common video compression techniques can significantly degrade performance by suppressing subtle traces of the generation process—an issue that affects many image-based detectors. Additionally, such detectors are inherently limited in that they cannot capture temporal cues or generation artifacts related to object or scene motion in video. Promising directions for improving video detection include incorporating a wider range of compression operations during training and developing more advanced entropy-based features that also account for temporal information. We plan to investigate these aspects more thoroughly in future work. Thank you for your understanding.

---

### Official Review · Reviewer_UjYv · 2025-07-03

**Clarity:** 4
**Significance:** 3
**Originality:** 3
**Rating:** 4
**Confidence:** 4

**Summary:**

Recently a various of techniques have been proposed for AI-generated image (AIGI) detection. Many of these approaches rely on spatial-domain cues, frequency-domain cues, or representations from pretrained models. Such methods often lack robustness when applied across diverse generative models and content types. This work introduces a novel approach to mitigate potential biases from model- and content-specific priors by proposing Multi-granularity Local Entropy Patterns (MLEP), which involves computing entropy feature maps across small patches over multiple image. After applying patch shuffling, multi-scale resampling and computing MLEP, a CNN-based classifier is trained for detection. The evaluation experiments demonstrates superior robustness and performance compared to other methods like FatFormer [1, 2]. These findings suggest that MLEP provides a content-agnostic and model-agnostic framework that substantially advances the state of AIGI detection.

*References*

[1] H. Liu, Z. Tan, C. Tan, Y. Wei, J. Wang, and Y. Zhao, “Forgery-aware Adaptive Transformer for Generalizable Synthetic Image Detection,” in *Proceedings of the IEEE/CVF Conference on Computer Vision and Pattern Recognition*, 2024, pp. 10 770–10 780.

[2] C. Tan, Y. Zhao, S. Wei, G. Gu, P. Liu, and Y. Wei, “Rethinking the Up-Sampling Operations in CNN-Based Generative Network for Generalizable Deepfake Detection,” in *Proceedings of the IEEE/CVF conference on computer vision and pattern recognition*, 2024, pp. 28 130–28 139.

**Questions:**

1. While the authors claim that real images exhibit a higher likelihood of reaching the maximum entropy value (i.e., 2.0), the magnitude of this difference varies significantly across models. Moreover, the distributions at lower entropy levels appear noisy and inconsistent. A more systematic or statistical comparison might strengthen the argument.
2. While the paper reports strong performance on several generative models such as StyleGAN2 and Stable Diffusion, it remains unclear how well the proposed method generalizes to truly unseen generative models. Additional evaluation on out-of-distribution generators would further strengthen the claims of generalizability.

**Ethical Concerns:**

["NO or VERY MINOR ethics concerns only"]

**Final Justification:**

Thanks for the rebuttal. My concerns have now been addressed.

**Limitations:**

yes

**Paper Formatting Concerns:**

no major formatting issues

**Quality:**

4

**Strengths And Weaknesses:**

**Strengths**:

1. The idea of this paper is novel and compelling. By computing entropy within small patches, the proposed MLEP framework captures fine-grained pixel-level differences, introducing a new metric for evaluating AI-generated image (AIGI) detection.
2. The authors conducted comprehensive experiments on GAN-based models (Table 1) and Diffusion-based models (Table 2), demonstrating the generalization ability of their method.
3. A thorough ablation study is presented, covering the three key components of the framework (Table 4), the impact of different resampling interpolation methods (Table 6), hyperparameter (Tables 5 and 7), and the effect of different classifier backbones (Table 8).
4. This paper is well-written and easy-to-follow overall.

**Weaknesses**:

1. The computational cost increases rapidly with larger patch sizes, which limits the scalability of the proposed entropy computation method to larger receptive fields.
2. Due to the small receptive field, the 2×2 patch may fail to capture broader contextual forgery cues, such as global texture inconsistencies or high-level structural artifacts.

---

> ### Author Rebuttal · Authors · 2025-07-27
>
> In response to **Question 1**, we would like to clarify that the entropy comparison shown in Fig. 1 of the manuscript between real and AI-generated images (AIGI) represents an initial observation in our study. It reveals that real images consistently exhibit a higher likelihood of containing 2.0-entropy blocks, suggesting they preserve finer local details compared to AIGI. This aligns with our hypothesis that AI-generated images tend to exhibit greater randomness and less structured detail as quantified by entropy. Motivated by this finding, we further explored the potential of local entropy as a discriminative cue for AIGI detection.
>
> In response to **Weakness 1**, we acknowledge the scalability limitation of the 2×2 window size. However, this configuration provides maximal computational efficiency, as a 2×2 patch has only five possible entropy states based on pixel occurrence patterns. This allows us to implement a fast, fully vectorized entropy computation method using PyTorch. Specifically, we extract all overlapping 2×2 patches via the unfold function, reshape them into flat vectors, and apply tensor operations to count unique pixel values and assign entropy values through broadcasting or indexing—without any explicit for loops. This approach enables parallel processing across all patches and is highly efficient, particularly when deployed on GPUs. The Python code used for local entropy computation is included in the supplementary materials of our initial submission.
>
> Regarding the concern about using 2×2 local entropy (**Weakness 2**), we emphasize that the core idea of our method is to eliminate high-level image semantics by scrambling small image patches. This design choice intentionally avoids reliance on semantic or content-level information, which may introduce bias, and instead encourages the model to focus on lower-level entropy distributions—capturing information randomness both within local patches and across randomly positioned pixel groups. To complement the 2×2 setting, we further introduce a multi-scale resampling mechanism, which effectively extends the entropy estimation to smaller image scales and enhances detection robustness.
>
>
> Finally, concerning the out-of-distribution evaluation (**Question 2**), we would like to confirm that our experiments were indeed conducted in a cross-generator setting. Our model was trained exclusively on ProGAN-generated images, and tested on images from a range of previously unseen GAN-based and Diffusion model-based generators, thereby validating its generalization across diverse generation methods. To further address your concern, we conducted additional evaluations using image samples generated by previously unseen models, including those produced by GPT-4o, Flux, Stable Diffusion, etc. Specifically, each set of generated images were paired with comparable number of real image samples selected at random, and we tested the model originally trained on ProGAN against these newly introduced datasets. The mean detection accuracy (Acc.)/average precision (A.P.) are reported in the following table, in comparison with the previous SOTA method, i.e. NPR (CVPR 2024). As shown, our method maintains consistently high performance across all these novel datasets, demonstrating its strong generalization ability in out-of-distribution scenarios.
>
> | Dataset | MLEP (Ours) | NPR (CVPR 2024) |
> |:------:|:------:|:------:|
> | GPT-4o | **100.0% / 100.0** | 100.0% / 100.0% |
> | Flux | 84.8% / 96.7% | **94.2% / 98.9%** |
> | firefly (from Synthbuster) | 93.3% / 97.9% | **93.4% / 99.1%** |
> | sdxl (from Synthbuster) | **99.9% / 100.0%** | 99.9% / 100.0% |
> | Midjourney-v5 (from Synthbuster) | 98.7% / **100.0%** | **99.4%** / 99.8% |
> | SD v1.4 (from GenImage) | **98.6% / 99.9%** | 92.3% / 95.6% |
> | SD v1.5 (from GenImage) | **98.7% / 99.8%** | 92.0% / 95.5% |
> | Wukong (from GenImage) | **98.6% / 99.9%** | 87.9% / 92.8% |
> | Mean | **96.6% / 99.3%** | 94.9% / 97.7% |
>
> > The two values in each cell of the above table denote Acc./A.P. respectively.

---

> > ### Comment · Reviewer_UjYv · 2025-08-04
> >
> > Thanks for the rebuttal. My concerns have now been addressed. I am still open to discussion before the final justification deadline.

---

> > > ### Author Response · Authors · 2025-08-07
> > >
> > > Many thanks for your thorough review and kind support.

---

### Comment · Area_Chair_UCyU · 2025-08-06
**[General Reminder for Authors and Reviewers] Author-Reviewer Discussion Phase Ending Soon**

Dear Authors and Reviewers,

As you know, the deadline for author-reviewer discussions has been extended to August 8. If you haven’t done so already, please ensure there are sufficient discussions for both the submission and the rebuttal.

Reviewers, please make sure you complete the mandatory acknowledgment **AND** respond to the authors’ rebuttal, as requested in the email from the program chairs.

Authors, if you feel that any results need to be discussed and clarified, please notify the reviewer. Be concise about the issue you want to discuss.

Your AC

---

### Note · Authors · 2025-08-14

In this final remark, we summarize the key strengths of our paper recognized by the reviewers, the major concerns they raised, and the actions we took during the rebuttal to address them. We believe this will help facilitate the ACs’ and SACs’ decision-making process.

All reviewers acknowledged the research motivation and novelty of the paper. Most agreed that leveraging entropy as a cue for AI-generated image (AIGI) detection is both reasonable and compelling, and that this approach clearly distinguishes itself from existing methods such as frequency-based analysis and CLIP-based models. Furthermore, the proposed method achieves superior performance compared to most baselines on standard benchmarks, a strength that was recognized by the majority of reviewers.

Reviewers raised several issues unrelated to novelty, including the method’s generalization to more generative models, robustness to common image processing operations, and the influence of generation models’ text prompts. In response, we conducted additional experiments on datasets covering diverse models and prompts, and compared against newer competitors (CVPR 2025 and ICCV 2025). Results consistently showed strong generalization in most cases, as acknowledged by most reviewers in their rebuttal feedback. Admittedly, our method still has some shortcomings, such as limited robustness to JPEG compression; however, we provided detailed reasoning and outlined potential solutions as future work. This was recognized by Reviewer EVyz, who raised their score from 3 (negative) to 4 (positive).

Overall, we are pleased to see that most reviewers provided positive feedback on our rebuttal, noting that their concerns have been addressed. However, we have noticed that Reviewer 3jko has not yet submitted the final Mandatory Acknowledgment, and we hope that our final remarks (or a reminder from the ACs) can prompt him/her to do so.

Finally, we sincerely thank all ACs, SACs, and reviewers for their efforts in organizing and conducting the review process. The reviewers’ comments have provided valuable suggestions and insights for improving our method and paper. If the paper is fortunately accepted, we will incorporate additional experimental results and further discussions in the final manuscript in response to the reviewers’ feedback.

---

### Decision · Program_Chairs · 2025-09-17

**Decision:**

Accept (poster)

**Comment:**

The recommendation is based on the reviewers' comments, the area chair's evaluation, and the author-reviewer discussion.

This paper studies the use of multi-granularity local entropy as features to train a CNN for classifying real v.s. AI-generated images. All reviewers find the studied setting novel and the results provide new insights. The authors’ rebuttal has successfully addressed the major concerns of reviewers. In the post-rebuttal phase, all reviewers were satisfied with the authors’ responses and agreed on the decision of acceptance.

Overall, I recommend acceptance of this submission. I also expect the authors to include the new results and suggested changes during the rebuttal phase in the final version.